# Whole-Body MRI in Rheumatology: Major Advances and Future Perspectives

**DOI:** 10.3390/diagnostics11101770

**Published:** 2021-09-26

**Authors:** Luca Deplano, Matteo Piga, Michele Porcu, Alessandro Stecco, Jasjit S. Suri, Lorenzo Mannelli, Alberto Cauli, Alessandro Carriero, Luca Saba

**Affiliations:** 1Department of Radiology, University Hospital of Cagliari, 09100 Cagliari, Italy; luca.deplano93@yahoo.it (L.D.); lucasabamd@gmail.com (L.S.); 2Department of Rheumatology, University Hospital of Cagliari, 09100 Cagliari, Italy; matteopiga@unica.it (M.P.); cauli@unica.it (A.C.); 3Department of Radiology, “Maggiore della Carità” Hospital, University of Piemonte Orientale (UPO), Via Solaroli 17, 28100 Novara, Italy; alessandro.stecco@uniupo.it (A.S.); profcarriero@virgilio.it (A.C.); 4Stroke Monitoring and Diagnostic Division, AtheroPoint™, Roseville, CA 95661, USA; jsuri@comcast.net; 5Department of Radiology, SDN, 80131 Napoli, Italy; mannellilorenzo@yahoo.it

**Keywords:** whole-body imaging, arthritis, bone, muscles, joint, rheumatology, pediatrics

## Abstract

Whole-body magnetic resonance imaging is constantly gaining more importance in rheumatology, particularly for what concerns the diagnosis, follow-up, and treatment response evaluation. Initially applied principally for the study of ankylosing spondylitis, in the last years, its use has been extended to several other rheumatic diseases. Particularly in the pediatric population, WB-MRI is rapidly becoming the gold-standard technique for the diagnosis and follow-up of both chronic recurrent multifocal osteomyelitis and juvenile spondyloarthritis. In this review, we analyze the benefits and limits of this technique as well as possible future applications.

## 1. Introduction

Whole-body magnetic resonance imaging (WB-MRI) consists of a series of high-spatial resolution images obtained in small body segments, with a field of view (FOV) of between 25 and 50 cm, and is combined by a dedicated reconstruction software that creates a whole-body image [1].

WB-MRI was introduced in clinical practice twenty years ago [2] and it was firstly applied in the oncology field for the assessment of skeletal involvement in many pathological conditions such as lymphoma, myeloma, and solid tumor metastases, providing good results similar to those obtained with bone scintigraphy, but with the benefit of being a radiation-free technique [3]. In the following years, technical improvements led to increasing sensibility for the detection of inflammation, allowing WB-MRI to be used in many other areas. Nowadays, it is largely used in the rheumatology field for the evaluation of the initial phases of diseases, for follow-up, and for treatments response [2].

In the past years, MRI in rheumatology was used for the analysis of single body regions by adopting a small field-of-view, with a high risk of losing other possible localizations of disease, especially the asymptomatic ones [2]. We can now consider WB-MRI as the gold-standard technique for the study of rheumatic diseases in pediatric patients due to its high-grade sensibility and lack of ionizing radiations [4]. Moreover, WB-MRI allows for visualization with a high-grade sensibility to contrast the inflammation of several districts such as tendons, muscles, bones, joints, and enthesis, providing a global view of the disease activity in the whole body; for these reasons, WB-MRI is now considered a fundamental technique in rheumatology [5].

The rapid evolution of sequences (in particular, the decreased amount of time that is necessary for these to be applied) has led the WB-MRI to be actually indicated as a non-invasive method in several musculoskeletal diseases, leading to questions regarding whether the old radiant methods have become obsolete in several situations [6].

## 2. WB-MRI: Acquisition Techniques

In recent years, the MRI FOV covers an area of 15–40 cm along the *z*-axis (longitudinal view) and this is allowed in a single session to study only a small part of the areas affected by any pathological processes [7]. Due to the improvement of hardware and sequences, it was possible to expand the FOV so that now we are able to cover the whole body with a single scan (Figure 1) [8]. Arms are the only portion of the body that often stay out of the FOV, thus needing a successive and separate scan for its study [9].

Short tau inversion recovery (STIR) and T1W are the basic sequences in the WB-MRI. STIR, also known as short TI inversion recovery, provides an overall view of the inflammation present in tendons, bones, joints, and enthesis [10]. The STIR is a fat suppression technique with an inversion time: TI = ln(2)·T1_fat_. In applying a 180° inversion pulse, fat tissues recover its longitudinal magnetization faster than water, crossing the 0 line (transversal magnetization) after a certain amount of time, which is called the “inversion time”. The application of another 90° excitation pulse in this period provides an image in which the fat signal is suppressed, thus there would not be any representation of the fat tissue in the final image [11]; it is usually acquired in coronal planes (Figure 2). Conversely, the T1 sequence offers better anatomic detail and allows for the identification of both disease-related structural changes and bone edema [10].

Other planes of acquisition, such as the sagittal for the study of the spine or the coronal for the study of the sacroiliac joint, can be used as smaller FOV-complementary exams depending on the specific disease or portion that has to be studied [2].

There was a discussion among experts on choosing the optimal magnetic field to use. Various sequences like T1,T2, STIR, and duffusion weighted imaging (DWI) have been extensively studied in the 1.5 T and it has been observed that the transition to 3 T devices has brought several advantages such as an increase in signal intensity and spatial resolution, with an associated reduction of acquisition times. The disadvantages of 3 T are mainly linked to an increase in chemical shift as well as to magnetic susceptibility artifacts, absorbed energy, and induced currents [12]. However, most of these limitations have been overcome and now WB-MRI is also possible with 3 T [13].

## 3. WB-MRI: Indications in Rheumatology

WB-MRI is one of the first-choice methods for both SAPHO syndrome (synovitis, acne, pustulosis, hyperostosis, and osteitis) and its juvenile counterpart CRMO (chronic recurrent multifocal osteomyelitis) [3,4]. More recent indications include soft-tissues and muscle multifocal inflammatory diseases as well as multiple sclerosis (Table 1).

### 3.1. Seronegative Spondyloarthritis

This group includes ankylosing spondylitis (AS), psoriatic arthritis, reactive arthritis, and the enteropathic arthropathies.

#### 3.1.1. Ankylosing Spondylitis

This is most characteristic among the seronegative spondyloarthritis, mainly affects young adults, and has a strong association with HLA-B27 [2]. Typical radiologic features include bone erosions, bone sclerosis, and progressive evolution towards ankylosis, which are mainly localized in sacroiliac and spinal fibrocartilaginous joints, with the possibility of peripherical enthesitis [2]. The poor utility of the conventional radiologic techniques and the low specificity of the symptoms often delay AS diagnosis; this also means a delay in the use of new and effective drugs (TNF-alpha inhibitors) that need specific radiological alterations in order to be administered [14]. Due to the ability of having a full view of the axial skeleton (the main site involved in the disease), MRI is able to detect sacroiliitis and spondylitis in very early phases, providing the possibility to start therapy as soon as possible [15]. As abovementioned, MRI is able to highlight pre-structural (subclinical) alterations such as bone edema and structural alterations previously seen. Inflammation of the sub-chondral marrow bone underneath the sacroiliac joint (BME) is actually included in the Assessment of SpondyloArthritis International Society (ASAS) classification criteria for axial spondyloarthritis [16]. Actual recommendations indicate the principal utility of SIJ (sacroiliac joint) MRI instead of whole rachis MRI [17].

ASAS classification criteria for axial spondyloarthritis include the presence of an area of BME in two consecutive sections or two distinct areas of BME in the same section of the sacroiliac joint [18], as described by the “Outcome Measures in Rheumatoid Arthritis Clinical Trials (OMERACT) [19]. According to the most recent recommendations, other inflammatory processes such as enthesitis and synovitis are not to be used as diagnosis criteria [17].

WB-MRI, by acquiring the entire skeleton and discriminating active lesions with chronic ones, allows to investigate the whole body and to find subclinical lesions, particularly in the dorsal rachis, significantly increasing the diagnostic confidence. The involvement of the posterior longitudinal ligament and the costovertebral joints is highly suggestive for AS [14]. The chest wall is another important area, particularly in regard to costochondral joints, sternoclavicular joints, and the manubrium-sternal joint. Extra-axial alterations such as synovitis and enthesitis can be observed in about 77% of the patients involved [14]. Unfortunately, upper-limb joints are almost always excluded from the FOV; futures technical improvements may be able to resolve this problem.

In conclusion, WB-MRI is actually fundamental for an early diagnosis and for identification of disease-activity areas in patients with chronic AS in order to determine the most appropriate therapeutic choice [20].

The study protocol should involve coronal STIR and T1 images of the whole body; sagittal STIR and T1 images of the whole spine; and oblique coronal STIR and T1 images for the evaluation of the SIJ. The DWI sequence can be used for a better view of the active inflammatory areas [21].

In clinical practice, replacement of the standard MRI with WB-MRI is difficult to accomplish due to two principal reasons: WB-MRI requires a long time to execute the scan (60/30 vs. 20/15 min) and both the rachis and SIJ MRI allows to perform an almost certain diagnosis of AS by themselves [10].

It has to be said that almost 20–50% of patients with negative SIJ MRI have extra-axial lesions [22].

There are four indications for the use of WB-MRI in AS [2]:Second-line exam in patients with unsure AS diagnosis after SIJ and rachis MRI as well as following strong clinical suspicion;Complimentary examination for advanced axial disease (ankylosis) to detect peripheral zones of active inflammation;Analysis of the extra-axial skeleton in seronegative spondyloarthritis with prevalent peripheral involvement; andEvaluation of the therapy response in the three cases just described.

#### 3.1.2. Juvenile Spondyloarthritis

Unlike the classic form, this mainly involves the peripheral joints and the enthesis; the axial skeleton is often spared at the onset of the disease but will be involved by the inflammatory process in the course of 5–10 years (about 40% of patients will develop a classic AS). The most involved sites are the lower-limb joints. There is a strong association with class 1 MHC, such as regarding HLA B-27 (present in 60–80% of patients). This disease affects about 20% of the children who suffer from chronic arthritis [23]. In using WB-MRI, we can observe the presence of inflammation not only in the enthesis but in all the structures associated with them, such as the fibrocartilage, bursae, and fat pads [6]. Enthesitis can be found in 60–80% of children who suffer from spondylarthritis. The most frequently involved sites are the patella inferior pole (50%), calcaneal insertion of the plantar fascia (38%), head of the metatarsus (22%), and calcaneal insertion of the Achilles tendon (22%) [6]. The edema of the bone marrow in the proximity of the enthesis is the most common MRI characteristic of juvenile spondylarthritis; however, it has low specificity considering it has been demonstrated that nearly 56% of children who suffer from spondylarthritis may present edema of the carpal bones. Thus, the presence of edema alone is not enough to make a diagnosis but must be associated to the clinical context [24]. The grade of the disease activity and the presence of unfavorable prognostic factors (such as the presence of sacroiliitis) are key factors in the choice of the correct treatment, particularly regarding the possible use of expensive drugs such as monoclonal antibodies. With the support of more studies, WB-MRI will become an important tool for early diagnosis, for the monitoring of disease activity, and for deciding whether to continue anti-arthritic treatment.

#### 3.1.3. Psoriatic Arthritis

MRI is much more efficient than the clinical or radiographic exam in the detection of synovitis and enthesitis, characteristics that are typical of this form [25]. As for AS, WB-MRI might play a role in monitoring the disease activity and in guiding therapeutic decisions.

#### 3.1.4. Multifocal Aseptic Musculoskeletal Disorders

SAPHO

This is a clinical syndrome characterized by non-septic osteitis and joint inflammation associated with skin lesions, with a peak of incidence in patients who are between 30 and 40 years old.

Every skeletal segment can be involved by the disease; the most common are the sacroiliac and sterno-clavear regions (70–90%) as well as the anterior chest wall (70–90%) [26]. MRI shows different characteristics depending on the age and grade of activity of the lesions.

In the active phase of the disease, focal bone lesions with bone edema can be detected, which are frequently associated with periosteal reaction. During the chronic phase, sclerosis and adipose involution are the most common discoveries. The observation of both active and quiescent lesions, along with their pattern of distribution, provides important help in terms of the differential diagnosis between SAPHO and other diseases such as tumors or multifocal infective bone processes (osteomyelitis, Ewing’s sarcoma, Langerhans cell histiocytosis, osteosarcoma, and leukemia) [27].

WB-MRI helps in early diagnosis by locating multifocal osteitis that interests even the extra-axial skeleton, allowing prompt treatments to take place that prevent several complications such as flat vertebrae and kyphosis in children [5]. It is also extremely useful to organize a follow-up and to evaluate the treatment response; all these merits establish this method as preferable in children instead of bone scintigraphy, particularly because of the absence of ionizing radiations. Moreover, bone scintigraphy is not always able to detect juxtaphysarian lesions that are often hidden by the physiological uptake of the nearby physis [28].

CRMO

This is a syndrome characterized by aseptic chronic inflammation that determines fluctuating pain in association with multifocal bone lesions. Most of the patients with this syndrome are children or adolescents.

The diagnosis can be difficult because symptoms, lab analyses, and X-rays are not specific to the disease. Even in this case, WB-MRI offers a better quality of images than scintigraphy and X-rays, and again it is a radiation-free technique [4]. The areas that are most commonly involved include the long-bones metaphysis, the hip bone, and the rachis (vertebrae plana is a common discovery in this scenario).

The remittent-recurrent trend of the lesions makes it difficult to perform a differential diagnosis with septic or neoplastic lesions; a biopsy is often necessary to perform a correct classification.

During the follow-up, WB-MRI can detect the persistence of active inflammation even in those lesions that appear to be in clinical remission [4].

### 3.2. Systemic Sclerosis

Among all the systemic rheumatological diseases, systemic sclerosis (SSc) is one of the most severe: it is characterized by a particularly intense inflammation followed by fibrosis phenomena that involves not only the skin and the subcutaneous tissue but even several internal organs such as the esophagus, lungs, and kidneys, leading to the disfunction and insufficiency of them in the worst-case scenario. Tendons, joints, and muscles are involved in 40–80% of patients with systemic sclerosis. Early identification of tendon and joint involvement can identify a subgroup of patients that have a higher risk of disease progression; this characteristic can be clinically difficult to discover or can be underestimated, a fact that establishes a potential indication for the use of WB-MRI for the locating of synovitis and tenosynovitis [29]. WB-MRI seems to be promising for the detection of the musculoskeletal involvement of the disease, which often appears symmetrical and multifocal, and is present in more than 50% of the patients (synovitis, tenosynovitis, myositis, enthesitis, and fascitis) [2]. MRI characteristics of skin-localized active inflammation (scleroderma) include:Hypointensity in T1-weighted sequences, along with the thickening of the cutaneous and subcutaneous plan; andHyperintensity in STIR sequences and gain of contrast in T1-weighted sequences after gadolinium.

MRI in systemic sclerosis is useful to identify the subgroup of patients with musculoskeletal lesions that have a higher risk of disease progression and organ damage. Another indication (actually under study) is the therapeutic follow-up to evaluate the changes the treatment induced in the disease course [2].

### 3.3. Polymyalgia Rheumatica

As a clinically diagnosed disease, characterized by pain and muscular weakness of the pelvic and shoulder girdle, Polymyalgia Rheumatica is associated with a moderate inflammatory state that involves all the joint structures and the nearby tissues (bursitis and tendinitis). Symptoms generally regress after the use of corticosteroids. WB-MRI can be helpful to exclude other rheumatic diseases, making clinical diagnosis easier [30]. Alterations in the extra-capsular signal, especially if located in the peri-acetabular space and underneath the pubic symphysis, are extremely specific characteristics of PMR [31]. This finding also correlates with good glucocorticoids treatment response [5].

### 3.4. Muscular Multifocal Inflammatory Diseases

This group includes idiopathic inflammatory myopathies and fascities. Among the first, they include polymyositis (PM), dermatomyositis (DM), and inclusion body myositis (IBM). PM and DM are both characterized by an autoimmune reaction directed against the striated muscle tissue and each have a similar clinical presentation (muscle fatigue, symmetry, and insidious onset). The muscle involvement can be isolated or associated with other diseases, especially those that affect the soft tissues such as in Sjogren syndrome, systemic lupus erythematosus (SLE), systemic sclerosis (SSc), rheumatoid arthritis, and antisynthetase syndrome (ASS). Patients with PM and DM generally present a proximal weakness of the limbs, an excess of fatigue, and a compromised muscular resistance. In addition to the striated muscle problems, patients may present difficulties in eating, swallowing, and breathing depending on which muscle groups are involved. In DM, there is also the possibility of cutaneous involvement that often manifests itself with the classic face heliotrope rash. PM, in contrast is not associated with the rash and is most commonly diagnosed in young adults. Lastly, IBM principally affects elderly men and women, and the muscular involvement is more often asymmetrical and distal [2,5].

#### 3.4.1. Antisynthetase Syndrome (ASS)

In this particular syndrome, muscular and breathing troubles are preceded by symptoms related to arthritis, synovitis, and tenosynovitis; that is why in the early phases of the disease, it’s common for it to be confused with rheumatoid arthritis [32]. In this case, WB-MRI can be extremely useful to detect both the muscular and extra-muscular involvement so that is possible to make a correct diagnosis [2]. WB-MRI allows for evaluating changes in both the signal intensity and thickness of muscular and peri-muscular tissue [33]. As for the signal, in active phases of the disease, it is possible to observe a hyperintensity in T1 and T2 STIR sequences post gadolinium, while there is hyperintensity in T1-weighted sequences in the chronic phase [2]. Changes in muscular thickness can consist of hypotrophic or pseudo-hypertrophic changes. WB-MRI is therefore able to confirm the muscular involvement and can lead the diagnostic process towards a specific phenotype according to the following.

The involved tissues including muscles, fasciae, and subcutaneous tissue. Muscular and subcutaneous tissue are both involved in DM and even the fasciae can be involved. In PM, we generally find isolated muscular edema.The topographical distribution of the lesions. In PM, shoulder and pelvic girdle are primarily involved.The portion of the muscle affected by the process (central, peripheric, or diffuse) [5].

#### 3.4.2. IBM

IBM shows a diffuse pattern [32]. WB-MRI is very helpful to detect active-inflammation portions of the muscle; this can show the most correct sites act which it is possible to perform a biopsy that will eventually confirm the diagnosis, avoiding “blindfold biopsies” that may lead to a false-negative result [2]. A biopsy of the muscular tissue is fundamental to assess the inflammatory nature of the disease and to determine if the lesion is in an active phase. It can also differentiate IBM from PM and DM, and can exclude other diseases such as metabolic myopathies or muscular dystrophies.

WB-MRI is also useful to diagnose the steroid-inducted myopathy, which is an entity characterized by adipose infiltration and muscular atrophy [5]. Lastly, WB-MRI represents an excellent method for the evaluation of the treatment response [34].

### 3.5. Neuromuscular Diseases

Signal characteristics are similar to the previous ones described and the WB-MRI determines which muscles or group of muscles are involved (examples of diseases are the Duchenne muscular dystrophy or the Becker muscular dystrophy). MRI-WB can be used in this group of disease to perform targeted biopsies and to assess the response to treatment [35].

### 3.6. Eosinophilic Fasciitis (Shulman’s Syndrome)

This disease is clinically characterized by painful symmetrical-hardened skin lesions (scleroderma-like) and the histology shows an inflammatory infiltrate composed of eosinophilic granulocytes and lymphocytes, with fibrosis and with the thickening of the fasciae [36]. WB-MRI can support the diagnosis, can guide the biopsy, and is useful during the follow-up and monitoring of the treatment response [2]. Typical forms show edema and the thickening of the fasciae in T2 STIR sequences, along with a notable gain of contrast in post-contrastographic T1-weighted sequences.

In less typical forms, there can be an involvement of the tissues close to the fascia, such as muscles and hypoderma [37].

### 3.7. Sarcoidosis

This is an inflammatory disease of unknown etiology that determines the formation of non-dairy granulomata in various body parts. WB-MRI is used to locate lesions in the spleen, lymph nodes, bones, and muscular tissue [38]. The multifocal involvement of the axial skeleton is very frequent and can be confused with bone metastases; the differential diagnosis process is aided by the fact that in sarcoidosis, there are lesions both in the active and quiescent phase (fatty lesions).

### 3.8. Langerhans Cell Histiocytosis

WB-MRI assumes charge over X-rays and bone scintigraphy in locating histicytosis lesions in children and eosinophilic granulomata in adults [5].

Findings primarily consist of multifocal areas of substitution of the marrow bone. Even in this case, diagnosis is performed due to the coexistence of active and quiescent lesions, with subsequent histological confirmation. WB-MRI is better than positron emission tomography (PET) in detecting lesions in the pediatric population, although PET appears to be more efficient during the treatment follow-up [39].

### 3.9. Avascular Multifocal Osteonecrosis (AVN)

This is a potentially disabling disease that can cause severe secondary arthropathy in the case of a late diagnosis. The lack of early symptomatology is often the cause of delays in both diagnosis and treatment. WB-MRI can detect ischemic lesions in epiphysis when they are in a subclinical phase; the lesions can be secondary to a vast group of causes, including lupus, hemoglobinopathies, high-dose steroid therapies, and oncological and rheumatic diseases [5]. MRI is considered the best method to detect the lesions that are characterized in T1-weighted sequences by multiple ischemic areas on the epiphysis which surrounded by a hypointense peripheric ring. Every site that may be involved in this process can be studied with a single WB-MRI scan [40].

### 3.10. Polyostotic Fibrous Dysplasia (PFD)

This is characterized by a non-neoplastic, multifocal proliferation of bone-fibrous tissue inside the bone marrow space. Usually, it is an asymptomatic disease but symptoms may occur in the case of pathological fractures, deformity, or when there is compression of adjacent structures [5]. The disease can be found in several syndromes such as McCune Albright syndrome (associated with precocious puberty and “cafe-au-lait” skin macules) and Mazabraud’s syndrome (associated with intramuscular myxomas). WB-MRI can evaluate the extension of the disease and can monitor symptoms’ appearance [41].

## 4. Conclusions

The principal role of WB-MRI in rheumatology is the study of seronegative spondyloarthritis, particularly the AS. Due to recent improvements, it is now possible to have a more efficient and accurate detection and stadiation of the disease. The more extensive FOVs allowed for the study of peripheral joints and the improvement in protocols provided both a new and important role to the diffusion-weighted sequences. WB-MRI is becoming the first choice in confirming the clinical suspect of SAPHO and CRMO, and for both the evaluation and follow-up of musculoskeletal-benign proliferative disorders such as the eosinophilic granuloma and sarcoidosis. The extensive range of WB-MRI can be helpful for diagnoses, for the evaluation of the amount of disease, and for biopsies and follow-ups during treatment of inflammatory conditions that affect muscles, fasciae, and subcutaneous tissues. Lastly, it can be applied in the follow-up of benign lesions that have a certain risk of malignant transformation, such as for multiple exostosis or nerve sheath tumors.

## Figures and Tables

**Figure 1 diagnostics-11-01770-f001:**
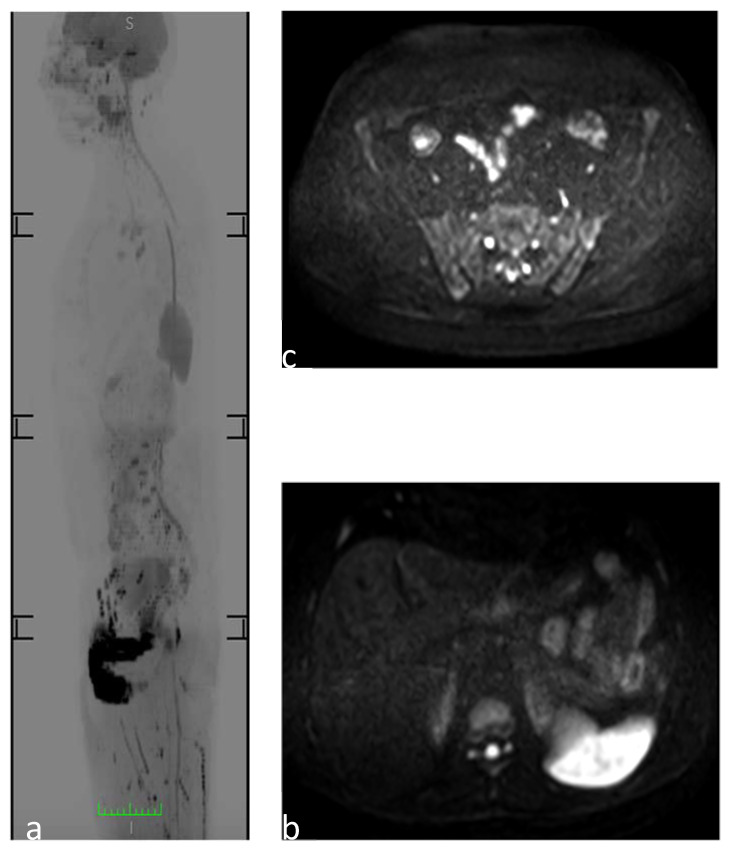
WB-MRI to study a 30-year-old man. (**a**) Sagittal reformatted diffusion-weighted image MR images (inverted gray scale, high b values, b 1/41,000 s/mm^2^) showing a physiological restriction to diffusion of the spleen and genitourinary system. (**b**) Axial DWI image at the level of the upper abdomen showing the physiological restriction to diffusion of the spleen and the “t2 shine through artifact” of the medullary canal. (**c**) Axial DWI image at the level of the sacroiliac joint, which does not show the pathological areas of restriction to diffusion of the osteoarticular structures.

**Figure 2 diagnostics-11-01770-f002:**
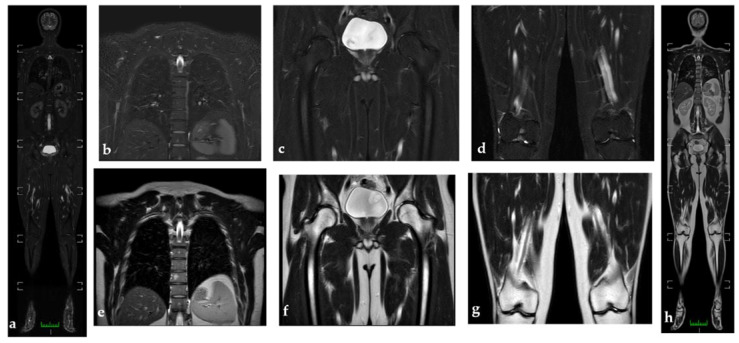
WB-MRI to study a 30-year-old man (**a**). WB-MRI coronal STIR images do not show active inflammation consisting of synovitis or tenosynovitis, and pathological findings appear to be better visualized than in WB-MRI coronal T2 (**h**). (**b**–**d**) represent coronal STIR images, while (**e**–**g**) represent coronal T2 images at the level of the thoracolumbar spine; the hip and knees do not show the pathological hyperintensities or hypointensities of the signal in the osteoarticular structures under examination. (**e**) Hyperintensity in t2 at the level of the soma of a thoracic vertebra with isointenity in stir (**b**); findings compatible with angioma.

**Table 1 diagnostics-11-01770-t001:** Highlights the osteoarticular sites that are mostly affected by the main rheumatological diseases, along with other areas less commonly involved or not belonging to the musculoskeletal system. Furthermore, the main tissue alterations caused by each pathology are described, visible on the WB-MRI as zones of altered signals [2].

Disease	Most Commonly Involved Areas	Other Areas	General Radiological Features	WB-MRI DWI
AS	Sacroiliac and discovertebral joints	Peripheral enthesitis and thoracic wall joints	BME, bone erosions, bone sclerosis, and ankylosis DWI is able to highlight active inflammation areas with high b-values	It has greater resolution power compared to STIR sequences in detecting inflammatory lesions and in distinguishing them from degenerative ones.
Juvenile spondyloarthritis	Peripheral joints (lower-limb joints) and the enthesis	Sacroiliac and discovertebral joints	BME in proximity of the enthesis	
Psoriatic arthritis	DIP and PIP joints (distal joints)	MCP/MTP and CMC/TMT joints (proximal joints)	Sinovitis and enthesitis; pencil-in-cup deformity	
SAPHO	Sacroiliac and sterno-clavear regions, and the anterior chest wall	Extra-axial skeleton	Presence of chronic (fibroadipose involution) and active (BME) lesions	
CRMO	Long-bones metaphysis, ankle, and calcaneus	Appendicular and axial skeletal	Non-specific signs of inflammation and relapsing-remitting lesions	DWI may be useful to distinguish malignancy from CRMO in the spine
Systemic sclerosis	Fingers, wrists, and ankles	Systemic disease: esophagus, skin, lungs, and kidneys	Synovitis, tenosynovitis, myositis, enthesitis, and fasciitis	
Polymyalgia rheumatica	Pelvic and shoulder girdle	NA	Inflammation of peri-acetabular space and underneath the pubic symphysis	
Polymyositis	Proximal limb muscles, symmetrical;	Swallowing and respiratory muscles	Inflammation of the affected muscles	
Dermatomyositis	proximal limb muscles and skin symmetrical;	Swallowing and respiratory muscles	Inflammation of the affected muscles	
IBM	distal muscles of the limbs, asymmetrical; and	Swallowing and respiratory muscles	Inflammation of the affected muscles	
ASS	joints, entheses, and synoviums	Respiratory and limbs muscles	Inflammation of these structures	
Eosinophilic fasciitis	Fasciae	Muscles and hypoderma close to the fascia	Inflammation with fibrosis and thickening of the fasciae	
Sarcoidosis	Multifocal involvement of the axial skeleton	Systemic disease: lungs, eyes, hepato-splenic, and muscles	Presence of chronic and active lesions	
Langerhans cell histiocytosis	Skull bones, upper limbs, and flat bones	Skin, endocrine system, and lungs	Coexistence of active and quiescent lesions	
AVN	Epiphysis long bones	Joints of the knees, shoulders, ankles, wrist, hips, and jaw	Ischemic lesions	
Hereditary ostechondromatosis	Flat bones or metaphysis of the long bones	NA	Multiple benign ostechondromas; signs of malignant transformation: growth of lesions after puberty or thickening of the cartilage hood	
PFD	There is no preferential bone location	NA	Multifocal benign proliferation of bone-fibrous tissue inside the bone marrow space	
Neurofibromatosis	Deformity of the orbit, facial bones, and spine	CNS and PNS	Nerve tumor that deforms adjacent structures	

## Data Availability

According to the typology of paper (narrative review) there is no data storage available.

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
