# Peer review of "Whole-Body MRI in Rheumatology: Major Advances and Future Perspectives"

_diagnostics, 2021, doi:10.3390/diagnostics11101770_

Round 1
Reviewer 1 Report
I read with interest this review performed by Deplano et al., in which authors reviewed the use whole body MRI in Rheumatology.
I have only minor comments.
On page 4, line 118, please explain the ASAS acronym.
I suggest to include a section analyzing the possible limitations of the use of whole body MRI in daily clinical practice.
Author Response
We would like to thank you and the reviewers for the time that was devoted to the accurate revision of our work, as it has definitely resulted in a more improved and complete article.
We tried to answer all of the questions and annotations we were asked to do, and we did our best to be as clear as possible by including all suggestions we received in the revised version of our article.
We hope all criticism raised have now been answered. Please do not hesitate to contact the corresponding author if any other changes to our Manuscript are necessary.
Looking forward to hearing from you soon, we send you our best regards.
Reviewer 1:
- Sentence : “On page 4, line 118, please explain the ASAS acronym.”
Reply: Dear Reviewer, thank you for the comment, we’ve proceeded to explain the ASAS acronym.
- Sentence : “I suggest to include a section analyzing the possible limitations of the use of whole body MRI in daily clinical practice.”
Reply: Dear Reviewer, thank you for the comment. Regarding the limitation issues about the use of WB-MRI in daily practice, we were unfortunately unable to add this particular topic to the final work.
We would like to thank you for the precious suggestion, that we will certainly consider for eventually future works.
Reviewer 2 Report
the manuscript addresses the usefulness of WB MRI in rheumatology
however, the manuscript consists in a listing of clinical and MRI features of various rheumatological disorders with little added value of WB MRI
several interesting questions related to WB MRI and rheumatology are not answered in the manuscript
- should WB MRI include the legs and feet? should it be different between children and adults as the covered area is smaller in children?
- how about the upper limbs and hands? for example in psoriatic arthritis or rheumatoid arthritis? should it be imaged or not?
- is contrast-enhanced fat-suppressed T1-weighted imaging useful to detect and quantify inflammation in WB MRI? is it more sensitive or more specific than fat-suppressed T2-weighted imaging or DWI?
- is there an added value to use Dixon sequences rather than fat saturated or STIR sequences? what about 3D WB MRI?
- what about the comparison between WB MRI and FDG-PET-CT?
Figures do not show any pathological findings - were there obtained in healthy volunteers? treated patients? Authors mention various rheumatologic disorders and describe their features at MRI, illustrations of some of the diseases would be nice
Author Response
We would like to thank you and the reviewers for the time that was devoted to the accurate revision of our work, as it has definitely resulted in a more improved and complete article.
We tried to answer all of the questions and annotations we were asked to do, and we did our best to be as clear as possible by including all suggestions we received in the revised version of our article.
We hope all criticism raised have now been answered. Please do not hesitate to contact the corresponding author if any other changes to our Manuscript are necessary.
Looking forward to hearing from you soon, we send you our best regards.
Reviewer 2:
- Sentence: “Figures do not show any pathological findings - were there obtained in healthy volunteers? treated patients? Authors mention various rheumatologic disorders and describe their features at MRI, illustrations of some of the diseases would be nice.”
Reply: Unfortunately, we don’t posses any WB-MRI image obtained in ours working hospital unit.
- Sentence: “What about the comparison between WB MRI and FDG-PET-CT?”
Reply: Dear Reviewer, thank you for the comment, we’ve not included a section about the comparison between WB-MRI and FDG-PET CT because this wasn’t the main purpose of this review. As a radiologists, we are also not qualified to the nuclear medicine field.
- Sentences: “Should WB MRI include the legs and feet? should it be different between children and adults as the covered area is smaller in children?”; “How about the upper limbs and hands? for example in psoriatic arthritis or rheumatoid arthritis? should it be imaged or not?”; “Is contrast-enhanced fat-suppressed T1-weighted imaging useful to detect and quantify inflammation in WB MRI? is it more sensitive or more specific than fat-suppressed T2-weighted imaging or DWI?” and “Is there an added value to use Dixon sequences rather than fat saturated or STIR sequences? what about 3D WB MRI”
Reply: Unfortunately, we didn’t have time to deepen the other topics.
Reviewer 3 Report
Nice review.
Please soften the statements regarding the use of WBMRI as gold standrd if no Evidence Based level of Evidence is reported (Line 97-98). This statement is inaccurate from the scietific point of view albeit referenced.
Delete the part related peripheral nerves, is too superficially described and not relevant for Rheumatology. The same for multlpe exostosis. The review is focused of Rheum not on WBMRI in general.
Author Response
We would like to thank you and the reviewers for the time that was devoted to the accurate revision of our work, as it has definitely resulted in a more improved and complete article.
We tried to answer all of the questions and annotations we were asked to do, and we did our best to be as clear as possible by including all suggestions we received in the revised version of our article.
We hope all criticism raised have now been answered. Please do not hesitate to contact the corresponding author if any other changes to our Manuscript are necessary.
Looking forward to hearing from you soon, we send you our best regards.
Reviewer 3:
- Sentence: “Please soften the statements regarding the use of WBMRI as gold standrd if no Evidence Based level of Evidence is reported (Line 97-98). This statement is inaccurate from the scietific point of view albeit referenced.”
Reply: Dear Reviewer, thank you for the comment, we’ve tried to change the statements about MRI being the gold standard technique.
- Sentence: “Delete the part related peripheral nerves, is too superficially described and not relevant for Rheumatology. The same for multlpe exostosis. The review is focused of Rheum not on WBMRI in general.”
Reply: Dear Reviewer, thank you for the comment. We’ve modified the part about the peripheral Nerves and multiple exostosis.
Round 2
Reviewer 3 Report
the manuscript has been sufficiently improved to warrant publication in Diagnostics